# Intensity of Multilingual Language Use Predicts Cognitive Performance in Some Multilingual Older Adults

**DOI:** 10.3390/brainsci8050092

**Published:** 2018-05-19

**Authors:** Anna Pot, Merel Keijzer, Kees de Bot

**Affiliations:** 1Department of Applied Linguistics, University of Groningen, 9712EK Groningen, The Netherlands; m.c.j.keijzer@rug.nl (M.K.); c.l.j.de.bot@rug.nl (K.d.B.); 2Unit for Language Facilitation and Empowerment, University of the Free State, Bloemfontein 9301, South Africa; 3Department of Applied Linguistics, University of Pannonia, 8200 Vezprem, Hungary

**Keywords:** hdl:10411/IHBPDH, multilingualism, cognitive control, inhibition, attention, older adults, language usage

## Abstract

Cognitive advantages for bilinguals have inconsistently been observed in different populations, with different operationalisations of bilingualism, cognitive performance, and the process by which language control transfers to cognitive control. This calls for studies investigating which aspects of multilingualism drive a cognitive advantage, in which populations and under which conditions. This study reports on two cognitive tasks coupled with an extensive background questionnaire on health, wellbeing, personality, language knowledge and language use, administered to 387 older adults in the northern Netherlands, a small but highly multilingual area. Using linear mixed effects regression modeling, we find that when different languages are used frequently in different contexts, enhanced attentional control is observed. Subsequently, a PLS regression model targeting also other influential factors yielded a two-component solution whereby only more sensitive measures of language proficiency and language usage in different social contexts were predictive of cognitive performance above and beyond the contribution of age, gender, income and education. We discuss these findings in light of previous studies that try to uncover more about the nature of bilingualism and the cognitive processes that may drive an advantage. With an unusually large sample size our study advocates for a move away from dichotomous, knowledge-based operationalisations of multilingualism and offers new insights for future studies at the individual level.

## 1. Introduction

Following research that has demonstrated that certain life-experiences can shape cognition (e.g., enhanced spatial memory in taxi drivers in London, jugglers, and musicians [1,2]), it seems intuitive that language, being one of the most intense and durable human life-experiences, could also enhance domain-general cognitive performance [3]. Cognitive advantages for bilinguals have indeed been observed in studies comparing performance of bilinguals and monolinguals on a series of cognitive tasks that measure (components of) executive control, most notably inhibition. Building upon the influential model of executive control by Miyake and colleagues [4] that distinguishes four components of executive functioning; inhibition, switching, monitoring and updating, the dominant view is that the enhanced cognitive performance of bilinguals is due to their continuous inhibition of the nontarget language in a specific context to resolve competition for selection, as both languages in a bilingual brain are always active [5]. This continuous cognitively effortful task would carry over into non-linguistic cognitive tasks, making bilinguals respond faster to non-verbal cues in especially conflict-monitoring tasks—such as the Simon, Stroop or Flanker task—than monolinguals [6].

A seminal paper by Bialystok and colleagues from 2004 demonstrated that there was a bilingual cognitive advantage (faster response on a Simon task) for older bilingual adults compared to their monolingual age peers and younger monolinguals [7]. In 2007, Bialystok and colleagues reported that bilingual patients diagnosed with probable Alzheimer’s disease received this diagnosis on average four years later than their monolingual peers, whereas they performed on a par on measures of cognitive control and there were no interfering effects of occupational level, gender and immigration status [8]. These findings collectively sparked a wealth of research on what has become known as the ‘bilingual advantage’ (BA), with varying results. Since 2004, the strength of a BA has decreased from strong to moderate effects for specific populations of bilinguals or no differences between bi- and monolinguals at all [9,10,11]. Recent (critical) reflections on the existence of bilingual advantages (see the discussion article by Paap et al. [12], and corresponding commentaries in Cortex) have given rise to calls to uncover more about the underlying constructs of language and cognitive control, in order to move our understanding of the differential results regarding a BA forward.

### 1.1. Cognitive Control

Hartsuiker [13] observes in *Cortex* that the research on BAs lacks clear theories on how language management influences cognitive control. He argues that we need information on the source domain (language control) and how this transfers to the target domain (cognitive control). Without being clear on the processes involved in the source and target domain, we cannot even begin to interpret the tranfer process. Currently, our main predictions on how bilingualism may impact cognitive performance is by the joint activation of languages in a bilingual brain, through which bilinguals exert enhanced control on processes of inhibition, monitoring or directing attention. This enhanced training carries over (it is unclear by which mechanism) into general processes of executive control.

However, there are a few problems with this view. First of all, the robustness of the inhibitory control account, whereby response inhibition was put forward as the driving force behind cognitive advantages, has been called into question by research on linguistic interference in (picture) naming. In a Dutch L1 picture naming experiment with a small group of university students, researchers found L2 English interference at the phonological level [14], suggesting facilitation and interference effects of the non-target language [15,16,17]. This goes against the notion that selection of the appropriate language is solely accounted for by the mechanism to inhibit the non-appropriate language (form), hence propelling researchers to argue that inhibition *alone* cannot explain executive control advantages (e.g., [10]).

Even more, studies demonstrate that the cognitive tasks used in behavioural studies to elicit a BA do not necessarily correlate with each other, revealing (different combinations of) a multitude of cognitive processes measured by these tasks (see [18]). Tasks that measure executive functions always tap into multiple components of cognitive control, creating ‘task impurity’ (see [19]). This makes it challenging to relate specific components of executive functions directly to bilingualism. Perhaps precisely because of this complexity and opaqueness, very few studies so far have attempted to grasp the underlying cognitive processes involved in bilingual decision making. This echoes the argument of Hartsuiker that clear theoretical underpinnings of the target domain and the transfer process are lacking.

### 1.2. What Is Bilingualism?

On the side of the source domain, in turn, we may have to take one step back and consider what it is that we define as bilingualism. Bilingualism is not a static ‘state’. Treating bilingualism as a dichotomous variable; solely as the ability to speak more than one language (the initial operationalisation in the 2004 study), falls short on acknowledging the vast differences between bilingual groups, or indeed individuals. Ihle et al. [20] tested whether the more languages an individual speaks, the better s/he performs on cognitive tasks, which would logically follow if the number of languages spoken directly relates to a BA. They conclude that, indeed, number of languages spoken contributes to cognitive reserve, yet not in all participants, and depending on other cognitively stimulating activities the participants engaged in, their verbal abilities in general, and basic cognitive processing speed. (See however Kave et al. [21] who longitudinally followed a group of multilingual elders in Israel and observed that the number of languages spoken reflects better cognitive performance, independent of educational level). A pure ‘knowledge-based’ (do you *know* multiple languages, yes/no) operationalisation of bilingualism therefore falls short of explaining differences in research towards a BA.

Perhaps the best way to solve some of the controversy in the BA debate is to stop comparing groups of mono- and bilinguals and rather pay closer attention to the type of bilingualism under investigation, treating this on a continuum based on bilingual language use rather than as a knowledge variable (also see [22,23]). This is not a new idea, yet something that has perhaps been overshadowed in recent years by a stronger focus on group-comparisons and dichotomous categorisations. In the 1960s, Cooper [24] demonstrated that Puerto-Rican speakers of English and Spanish in the USA performed differently on word naming and association tasks depending on how much they used each language in five societal domains; home, religion, neighbourhood, education and work. He advocates for more fine-grained operationalisations of bilingualism according to how the languages are used. Similarly, Grosjean [25] argues for the inclusion of language modes (how long is a subject in a monolingual or bilingual mode, and how much switching takes place in this bilingual mode), language stability and language function, and a more detailed account of the language history of subjects in bilingualism research.

The interactional nature of bilingualism, which goes beyond the static notions of number of languages and simultaneity of acquisition, is captured in Green and Abutalebi’s [26] adaptive control hypothesis (ACH). The ACH posits that different interactional contexts, either single-language, dual-language or dense code-switching contexts, place different cognitive demands on an individual (related to e.g., conflict monitoring, interference suppression and goal-orienting). Greater neural efficiency is expected to be observed for bilinguals who frequently reside in a dual-language context, as they show skills in monitoring (language) cues, allowing for more rapid switching. In dense code-switching contexts and single-language contexts, effective communication is less affected by careful background monitoring of language cues, suggesting fewer switching advantages for bilinguals in these contexts.

The differential contextual demands of diverse populations of bilinguals may be explanatory in understanding the presence or absence of bilingual advantages. An illustration of the ACH is a study by MacNamara and Conway [27] towards bimodal bilinguals (cross-sectionally measured with two groups of college students; one with and one without two years of experience with American sign language). They looked at whether high Bilingual Management Demands (BMDs), operationalised as degree and frequency of switching between languages (cognitive control), and the experience with managing those demands is the mechanism responsible for cognitive enhancement. High BMD experience was associated with better performance on tasks of cognitive control and working memory capacity after two years. They suggest that rapid switching and the coordination of simultaneously comprehending and producing in two languages, which becomes more efficient with experience, enhances cognitive control. Because of the variation within bilingual populations with regard to BMD (type, experience and magnitude), the presence and absence of a bilingual advantage also varies, they argue.

Bilingual advantages may then have more to do with the domain of switching between languages. Indeed, studies have found that the degree with which bilingual switch languages is predictive of a bilingual advantage [28,29,30]. However, language switching, especially when measured in isolation and under strict, artificial conditions (e.g., cued switching), is problematic from the view whereby language use is mostly interactional in nature. The question may be asked whether bilinguals indeed exert language control and inhibit one language when switching to the other, especially in conversational interactions where switching may occur mid-sentence, and the strict ‘boundaries’ that separate languages do not apply. Moreover, as de Bot (2017) aptly notes, switching is not unique to bilingual settings. Monolinguals may switch codes or registers depending on the situation. When switching is such a common phenomenon, both in bi- and monolinguals, can we than really assume that switching costs (even though evidence for lower switch costs for bilinguals is robust), or inhibition, is enough to drive a BA? (see [31]).

Perhaps it is not merely language control, but rather the environment in which language is used, possibly in addition to other cognitively enriching experiences, by which a cognitive advantage may be observed. Informative in this regard is a study on performance on EF tasks by bilinguals and two monolingual groups in different linguistic environments (French-dominant Quebec and English-French bilingual Ottawa). The language environment seemed to offer a more robust explanation for enhanced EF performance—by English monolinguals in Ottawa, where they are exposed to French in their environment, contrary to French monolinguals in Quebec, who lack this bilingual exposure—than bilingualism itself [32].

The influence exerted by the environment on experiments is often overlooked. With an analogy of the temperature of boiling water—which depends on altitude—Bak argues that even when we control for differences in experimental settings, the selection of participants, and different methods of data analysis, the environment in which an experiment is conducted may result in different observations [33]. As such, he advocates for the importance to compare results conducted in different environments, rather than replicating the same experiment in the same environment. As above, the context in which bilinguals use their languages (in a highly bilingual environment where switching is practised daily as opposed to a monolingual environment) may offer an explanation for some of the conflicting evidence for a BA.

Moreover, Bak argues that the attitudes towards bilingualism or certain languages in different environments (which may be more positive in highly bilingual populations such as in Brussels or more negative when language use is politically coloured) may additionally play a role in the conflicting evidence; believing that being bilingual is an asset rather than a disadvantage. A recent investigation into the different operationalistations of bilingualism in studies towards BAs between 2005 and 2015 [34] revealed that the degree to which different characteristics of bilingualism are reported differs greatly. Moreover, there is also a lack of sociolinguistic information, which is of particular importance when viewing bilingualism as an interactive life experience. The authors advocate for a better documentation of the social context (usage and status of languages in the population), but also the quality of foreign language instruction, by which in some countries bilingualism is not a life experience, but a learning experience.

In a recent contribution to a special issue of Linguistic Approaches to Bilingualism, Valian [35] argues that, as experiences accumulate over the lifespan, singling out executive function benefits belonging to certain experiences is increasingly difficult. Benefits of bilingualism may be additive, or additive up to a certain point or are only visible when they occur together with other benefits (e.g., from being physically active, playing a musical instrument, a specific diet, and so forth). The fact that there are no negative results reported for bilinguals (only positive or null results), leads Valian to propose that there is a benefit, but that this benefit competes with other benefits, hence showing positive results in some populations, but null results in others. Because individuals are so diverse in experiences, and tasks measure different aspects of executive functions, she argues that it is more likely that there are also different mechanisms by which executive functions may be enhanced.

When BA’s are so haphazardly observed, we might want to delve deeper into the construct of bilingualism. Bak [36] speaks of a forest of confounding variables in research on bilingualism. Hence, rather than a factor in isolation, bilingualism may contribute to enhanced cognitive performance when it is viewed together with other experiential factors. The visibility of such an effect then perhaps depends on the presence and strength of these other factors (see [37,38]). What we need are large-scale studies whereby bilingualism is present in many forms, and examine not only its static presence, but also consider language usage patterns, to be able to gauge whether there is indeed a benefit attributable to bilingualism, or whether BA’s lie in the combination of experiential factors. This study is one of the first attempts to investigate the nature of multilingualism and possible cognitive effects in a large and diverse group of older adults.

### 1.3. This Study

Moving away from a static, isolated and knowledge-based view of bilingualism, this paper details the observations of a study among a highly variable group of multilingual older adults in a small geographical area in the northern Netherlands, rich in dialects and languages. Given the contextual nature of bilingualism, it is informative to look at a bilingual population in an area where bilingual practices are widespread, but the degree of bilingual language use and type (two languages, dialect and standard language combinations, etc.) differ from person to person. We answer to the call to uncover more about the nature of bilingual advantages by investigating the different types of multilingualism (language combinations, usage intensities, social context, etc). By doing this in a population that is multilingual to varying degrees and in varying forms/manifestations, comparison to a monolingual population is not needed, and ineffective.

The main aim of this study is twofold:
To uncover more about if, and what aspect(s) of, multilingualism may facilitate enhancement of executive functions and what this tells us about the nature of multilingualism (as a knowledge- or experience-based variable) and the cognitive constructs that are involved;Whether multilingualism can contribute to enhanced cognitive performance in the presence of other (known) ‘confounding’ factors relating to health, wellbeing, and quality of life.


For sake of continuity based on the persistent ideas of cognitive control in the literature, we assess inhibition, attention-direction and set-shifting processes in a diverse elderly multilingual population. For the first question, we expect that it is not the number of languages or degree of proficiency of individuals (the knowledge-based operationalisation) that enhances cognitive performance, but rather the intensities with which different multilinguals (who may differ in number of languages, proficiency and language combinations) use their languages in different contexts. For this we draw on the premise of the adaptive control hypothesis, in that we expect that a more balanced use of different languages across different social contexts elicits better (faster or more accurate) performance on cognitive tasks related to switching and attention, similar to the observations by MacNamara and Conway [27]. If this is indeed the case, this may confirm the speculations that there are different BA’s for different populations, and that these populations likely mainly differ in language usage intensities.

However, to truly assess the uniqueness of bilingual populations, and whether populations with enhanced EF performance can effectively be discerned on the basis of their language usage the second question forms the basis of a model in which we insert the linguistic information of our participants, together with demographic, health and lifestyle information. We may be hard-pressed to find that speaking multiple languages is more effective for EF enhancement than other lifestyle factors, such as playing a musical instrument, and perhaps that only by putting these factors together in a model, a significant effect can be observed. Or there may be other, environmental factors, that offer a more ready explanation of enhanced EF performance. The great variety in our multilingual sample in terms of language experiences, usage intensities and other demographic, health and wellbeing characteristics offers a unique insight into how these factors may, or may not, interact with each other and perhaps, under specific circumstances, show an effect of multilingualism, in some type or form, on cognitive performance. At least it offers more insights into why some research does and other does not find bilingual advantages, while at the same time also demonstrating the complexity of studying one isolated factor (bilingualism) in a social context.

## 2. Methods and Materials

### 2.1. Participants

By means of calls put out in the local media (radio, newspapers) information flyers and personal networks we recruited participants in the three northern provinces of the Netherlands. Inclusion in the study was based on the following criteria:
Participants were 65 years or older (the benchmark at which many people in the Netherlands used to retire).Participants were residents of one of the three northern provinces in the Netherlands (Groningen, Friesland or Drenthe).Participants were cognitively ‘healthy’ individuals; i.e., they did not consciously suffer from any form of cognitive impairment. (This was asked in the questionnaire and people with an affirmative response were excluded).


A total of 387 participants took part in the study. Participants ranged in age from 65 to 95 years, with a mean of 72.07 (sd = 5.7). Participants were more or less evenly distributed among the three provinces and the dataset was nicely balanced in gender (201 male and 185 female participants (missing information of 1 participant)), also see Table 1.

From Table 1 it becomes apparent that, overall, the participant sample is relatively highly educated (mean of 5 (higher education) on a scale of 1 to 6) and have a moderate to high income (around 2500 euro gross per month). They are in good average health (a mean of 3, equivalent of “good health”) and rate their quality of life generally high, with an 8 on a scale of 1 (low) to 10 (high). Two-third of the participants lead an active lifestyle and around a third of the participants play or have played a musical instrument.

Ease of access to the study was facilitated by giving participants the opportunity to participate online from home (which automatically creates a bias by including only people proficient in handling computers). By means of the online software application Qualtrics [39] participants completed an extensive background questionnaire on demographics, health, wellbeing, personality, social networks and language proficiency and use. In addition, participants completed three cognitive tasks targeting executive functions. In addition to online participation, people without access to a computer could also set an appointment for a face-to-face interview at home. This resulted in about 90% online participation, and 10% home-visits. Both the online and face-to-face interviews adhered to the same setup, hence we did not expect any quality differences between the two modes of data collection.

Participants were informed about the nature of the project prior to the study and it was made explicit that by completing the questionnaire they gave active informed consent that their data may be used anonymously for scientific purposes (in compliance with the departmental guidelines for participant testing at the Faculty of Arts of the University of Groningen). Answers were anonymized by having participants create a unique ID-number based on a set of questions. This ID-number was created at the start of the questionnaire, and had to be filled in again (prompted by the same questions) at the start of the cognitive tasks. This way, data from one participant could be easily combined without the need to obtain personal identification information.

Not all participants filled in all questions or performed all cognitive tasks. A total of n=387 completed the language questionnaire. Of these 387 participants, n=311 also completed the first cognitive task (an Eriksen Flanker Task), n=292 completed a Wisconsin Card Sorting task and n=193 also completed a Corsi forward span task.

Overall, the participants were well-educated (M = 4.9 on a scale of 1–6, 6 being a university degree) and rated their quality of life generally high (a mean score of 8 (sd = 0.91) on a scale of 1 to 10). The group was relatively in good to average health (M = 3.3 on a scale of 1 to 5) and the majority was physically active. Almost half of the population played a musical instrument (around 48%). For the calculations, all available data was used, however, as one measure has more data points than another, results need to be interpreted with caution.

### 2.2. Multilingualism in the Northern Netherlands

For this study we consider bi- or multilingualism along a continuum, rather than dividing participants in groups based on the number of languages they speak or relative proficiency. Therefore, we explicitly informed participants that they could participate regardless of the number of languages they spoke and mastery of these languages. In addition, we stressed that we also regarded (regional) dialects also as linguistic varieties which can be listed separately from languages such as Dutch or German, as previous research has shown that dialects may also be regarded as ‘separate’ languages (Kirk et al. [40]). As the northern part of the Netherlands is rich in (regional) dialects, we hence defined multilingualism as any combination of languages or dialects. To uncover more about the operationalisation of multilingualism we included questions in the language questionnaire pertaining to:
Number of languages, and which, in order of dominance (maximum 5).Number of languages, and which, in order of acquisition (maximum 5) (these questions were based on the LEAP-Q questionnaire [16] (see below).Self-rated proficiency (speaking, reading, listening and writing) in the first three languages listed (5-point scale).Age of onset of acquisition of the first three languages and mode of acquisition (school, work environment, friends/family, or combinations).Relative usage intensity of the five languages (ranking from low to high).Degree of usage (5-point scale) of each language in different social domains (see Section 2.3.1)Switching behaviour measured with questions from the Bilingual Language Switching Questionnaire [41].Language attitude


Regarding languages, the group of participants was highly multilingual, with an average number of 4 languages and most individuals reporting to know no less than three languages. As is inherent to self-reported data, some individuals might have been more inclined to also list the languages they know but rarely use, such as school-languages (English, German and French), whereas others only listed those languages they actively use on a daily basis. The high number of reported languages also stems from the fact that we asked participants to list languages as well as dialects. See Table 2 below for the language details.

Proficiency in both the first and second listed languages of the participant is overall high (a mean close to the maximum of 5), and proficiency in the third language is on average slightly lower (but with more variation between participants considering the higher standard deviation of 0.8). The second and third language is acquired at a wide span of ages, but on average slightly later than the L1. Two third of the participants are categorised as “early bilinguals” according to whether they learned their second language before or after the age of 12. Participants generally have a positive attitude towards speaking both their first, second and third language, with high mean scores of around 4 on a 1–5 point scale. There is variability in the extent to which participants use their L1, L2 and L3 across different societal domains. The score is an aggregated total of use of each language in a specific social domain (see Section 2.3.1). The L1 is used across most domains (high mean score). The L2 and L3 are more tied to use in a number of specific domains, given the overall lower mean score. Degree of contextual switching is moderate.

### 2.3. The background questionnaire

A questionnaire targeting demographic, health, quality of life, personality and language information of the participants was distributed using the online questionnaire platform Qualtrics [39]. Participants were presented with a welcome screen which listed the different elements of the questionnaire and informed participants about the procedure, before continuing to the questions.

Table 3 below gives an overview of the different domains and questions asked in the background questionnaire.

The health and wellbeing questions were derived from a standardized questionnaire on health and quality of life (The Older Person and Informal Caregivers Survey (TOPICS)), that is used in numerous projects targeting older adults [42]. Questions and scoring methods can be downloaded from their website.

We tried to capture as many factors as possible that we deemed could contribute directly to cognitive performance in the background questionnaire. We selected those factors that have been shown to influence cognitive abilities in previous studies, such as an active and socially integrated lifestyle [44]. We therefore assessed measures of health, quality of life, wellbeing and hobbies (playing a musical instrument [45] and being physically active [46]). In addition, previous research has shown that a resilient personality profile (less affected by stress, anxiety and depression) is related to a delay in the onset of clinical dementia [47]. Also, low levels of conscientiousness and agreeableness and higher levels of neuroticism are associated with higher risk of cognitive impairment [48]. Therefore, we also assessed wellbeing (with the TOPICS-MDS) and compiled a personality profile for each participant with the TIPI personality scale [43].

We also asked participants to provide information on their primary social network (type and frequency of contact with five relations, and in which language this contact took place) as being socially active is related to less cognitive decline in old age [49]. However, due to the operationalisation of the question, quantifying this information proved too difficult. For the current study, therefore, we discarded this variable. In addition, due to time constraints and the multitude of confounding factors we did not take into account e.g., a healthy diet [50] or other leisure activities [51]. However, we do have more detailed information on the specific hobbies our participants engaged in (e.g., gardening, quilting, etc), which calls for a future study with more fine-grained analyses at the individual level.

#### 2.3.1. Data Transformation

In order to run statistical analyses on the questionnaire data, we collated the healthcare questionnaire from the TOPICS-MDS questions into two variables (functional health and emotional wellbeing) by means of a Principal Component Analysis (PCA) conducted in R [52]. Similarly, we did not include all questions from the original BSWQ into the questionnaire, through which we were unable to use the scoring method applied to the original BSWQ. By means of a PCA we extracted three meaningful variables: contextual switching, conscious switching and unintended switching. A resilience index was calculated by using the scoring procedure laid out in the TOPICS-MDS scoring instructions, by combining answers to functional and emotional wellbeing questions. Quality of life was determined by a self-rating on a scale of 1 (lowest) to 10 (highest).

Regarding the language measures, number of languages was simply a count variable. Type of language combinations was determined by selecting the first two languages of each participant and determining its origin; either Dutch, Frisian, regional dialect (Gronings or Drents), Germanic, Roman or ‘Other’ (more categories would make the dataset too fragmented). These combinations (e.g., Dutch-Frisian, or Regional dialect-Dutch) were subsequently given a number (1 to 9) and entered as a ‘factor’ variable into R. Early versus late bilingualism was calculated by considering the age of onset of acquisition of the second language of each participant. The cut-off point for a classification into the ‘early’ category was 12 years old, the age at which children in the Netherlands attend high school and structurally receive English, German and French language classes.

Intensity of language usage was regarded through the amount with which each language was used in different social domains (the family domain, friends, neighbours and acquaintances), measured on a 5-point scale variable (never to always). A composite score was calculated for each language (up to language three), in which the ratings for each language over the different domains are averaged to create one ‘usage’ score per language across different domains. A maximum score of 5 indicated that this language is always used across domains. Participants could give a maximum score of 5 (‘I always use this language in this domain’), hence the index variable is a score between 1 and 5.

Attitude towards the participant’s first three languages was measured by asking whether the participant liked to speak the language and deemed it important, on a 5 point scale. Finally, the five personality traits were calculated by adhering to the scoring procedures of the TIPI [43], whereby participants list to which degree (1–7 scale) they find certain traits applicable to themselves.

These variables were added to a dataframe in R, together with the results of the three cognitive tasks, resulting in 387 rows with data for all participants, with some missing values when participants did not complete (one of the) cognitive tasks. The dataset will be made available in DataverseNL under [53].

### 2.4. Cognitive Tasks

Participants completed three cognitive tests, administered via the freely online available program Psytoolkit [54], with which users can create their own reaction-time experiments using a simple scripting language. The advantage of using Psytoolkit over other digital programmes is mostly that Psytoolkit allows the user to distribute a task online (contrary to Eprime or Pebl). The downside of using Psytoolkit is that the key-logged response may be slightly delayed depending on a participant’s network strength, and participants may abort the task before finishing, which results in data loss (Psytoolkit only stores data from completed tasks).

Data from each participant were stored in separate .txt files (one per task) on a secure server. The data from the text files were subsequently imported into the statistical programming software RStudio [52]. From the raw datafiles, the desired measurements were calculated per participant and these were combined based on the participant’s ID number, so that each participant was a row in a dataframe with a Flanker effect score, a WCST persistent error score and a Corsi span score.

Table 4 below gives an overview of the cognitive tests that were used.

For this study, we only focus on performance on the Flanker and the WCST, as these two are most relevant for the questions asked in this paper.

#### 2.4.1. Flanker Task

The Flanker arrow task was adapted from the version listed in the experiment repository of Psytoolkit, based upon the original Flanker task by Eriksen, which uses letters as stimuli instead of arrows, as is the case here. Slight modifications to this experiment were made in the form of a more detailed set of instructions beforehand (in Dutch) and the use of bigger arrows (white arrows against a black background). The tasks consisted of in total 50 trials. Participants pressed a key on the left (Q) or right (P) of their keyboard depending on the direction the middle arrow (flanked by two on either side) is pointing in. The flanking arrows may point in the same direction as the middle arrow (congruent) or in the opposite direction (incongruent). Items were presented one by one in the middle of the screen and preceded by a fixation cross (300 ms). After participants completed 50 trials their reaction time per item, accuracy and congruency was saved. From these data a Flanker effect score was calculated by subtracting the mean reaction time on the congruent trials from the mean reaction time on the incongruent trials. A lower score is an indication of a lower cost in responding to the incongruent trials as opposed to congruent trials, reflecting either faster processing speed, enhanced attention or better inhibition.

Data-cleansing procedures may skew the data. In a brief report, Zhou and Krott [58] review the appearance of a bilingual advantage in reaction time experiments and demonstrate that those studies that do not apply data-cleansing procedures more often report a bilingual advantage than studies that trim the reaction time data to fall in between an accepted range. For this study we decided not to trim the data, yet the dataset reveals that the maximum reponse on the flanker task was below 1700 ms.

#### 2.4.2. Wisconsin Card Sorting Task

The Wisconsin Card sorting task requires participants to sort a set of cards (160 in total) according to a (changing) rule. Participants were presented with a screen with 4 cards, representing the sorting stacks. Each card showed a number of symbols (1 to 4 circles, triangles, stars or squares) in a particular color, and participants sorted each new card that appeared below these four stacks onto one of the stacks based on a rule: sort by color, shape or number. The rule changed with every ten sorted cards. Participants did not know the sorting rule beforehand, but received feedback after every card (correct or incorrectly sorted according to the rule). Through trial and error participants figure out the rule and sort each following card according to this rule. Until the rule changes. The positive feedback (Correct!) then suddenly changes to negative (Incorrect!), signaling that participants have to change the sorting rule. The WCST logged the number of times participants sorted a card correctly and, more importantly, whether they persistently sorted correctly (which indicates that they apply a rule). When participants persisted with applying the old rule, their number of persistent errors increased. A higher number of persistent errors reflects more difficulty in switching to a new rule (which may increase when participants have had to switch a number of times), reflecting the degree of prolonged attention and switching ability.

#### 2.4.3. Statistical Analyses

To answer the first research question regarding the underlying mechanisms of multilingualism, we fitted two multiple linear regression models with the same fixed variables—one for each cognitive task—using the R function ‘lm’. The dependent variable in each model was the outcome of the cognitive task. We computed a regression model first with only the ‘knowledge’ factors:
Number of languagesLanguage proficiencyEarly vs. late onset of acquisition of language 1 and 2Language combinations


Next, we fitted a regression analysis with only the ‘usage’ variables of multilingualism, and observed whether the model improved when we included an interaction of the two variables:
Across-domain usage of languages 1, 2 and 3Contextual language switching


The outcomes of these two measures were subsequently compared.

To examine the second question, namely which (combination of) variables contribute to enhanced cognitive performance as measured by shorter Flanker effects (incongruent – congruent RTs) and fewer persistent errors on the WCST, we analysed the data using a multivariate partial least squares (PLS) regression model (with the package “pls” in R).

PLS regression is especially useful in examining which combinations of variables (that might be correlated) explain the most percentage of variance, in our case in a model with Flanker RTs or WCST error scores as the predicted variable. Given our hypothesis that multilingualism is a contextually embedded variable—an experience—it is perhaps likely that there are other factors that covary with (aspects of) multilingualism. Rather than building a linear regression model, which typically assumes that all variables are independent and seeks which factor(s) in isolation explain part of the variance (as we did with the predictors of multilingualism above), PLS regression is a well-suited technique in instances where factors may be correlated to some degree.

We built two models, one with the Flanker effect scores as the predicted variable, and one with the WCST error scores as predictor. Our independent variables are the scores derived from the extensive background questionnaire (cf. Table 3), which included measures of health, quality of life, multilingualism and personality. A list of the factors that were included is presented below:
AgeGenderEducationIncomeSelf-reported healthQuality of LifePlaying a musical instrumentSports/being activeNumber of languagesProficiency in L1/L2/L3Age of onset of acquisition L1/L2/L3Early vs. late language acquisitionAttitude towards L1/L2/L3Degree of contextual switchingThe Big Five personality traitsAcross-domain usage of L1 (see Section 2.3.1)Across-domain usage of L2Across-domain usage of L3Language combinations (as factor with 9 levels (combinations of first two languages))


A PLS regression model tries to maximise the covariance between the dependent variable (Y) and the predictor variables (X). Hence, PLS selects components of X (similar to a principal component analysis) and computes the optimum number of predictors that are relevant in explaining Y (and thus explain the maximum covariance between X and Y) (see [59,60] for more information).

Our PLS regression models were first computed with the default 10 components, after which the output was examined and the minimum number of components was retained (when the CV value does not increase but levels off or decreases). By means of a Variable Importance in Projection (VIP) score for each variable we assess which variables provide a meaningful contribution to the model (greater than 1). A bar graph of the VIP scores was subsequently used to view each variable’s importance. With such a procedure, we prevented ‘gold-digging’ in the data (i.e., selecting significant variables by a degree of randomness). This is something we will return to in the discussion

## 3. Results

### 3.1. Descriptives

The statistical summary of the cognitive test scores in Table 5 lists mean performance on the two cognitive measures.

The Flanker effect score (the difference between congruent and incongruent trials) ranged from positive to negative results, indicating that whereas the majority of the participants were faster on the congruent trials (hence the positive mean score), some participants responded faster to the incongruent trials. This is striking, as there is apparently a negative Flanker effect. A histogram (Figure 1) shows the distribution of Flanker effect scores.

The mean number of persistent errors (how many times a participant continues applying the old rule when s/he has to switch to a new rule) in the WCST was similar to the average mean population score of 11 errors reported elsewhere in the literature [61], yet should be interpreted with caution. After the experiment, a number of participants reported that they experienced difficulty with the WCST. The task is relatively complex in comparison to the Flanker task, and when participants are not clear on the instructions they may just guess the correct rule most of the time. Therefore, the results obtained from the online WCST may be less reliable than the results for the Flanker task.

### 3.2. Static Variables of Multilingualism

We first ran two linear mixed effects regression models on the outcomes of the two cognitive tasks (Flanker effect score, WCST error score) with the static ‘knowledge’ factors of multilingualism as predictor variables: number of languages, early vs. late bilingualism, proficiency in language 1, 2 and 3, and the different language combinations (Dutch-Frisian, or dialect-Dutch, etc). The output summary is presented in Table 6 and shows two non-significant models (one model for each cognitive measure).

The table above demonstrates that the static operationalisations of multilingualism—as number of languages spoken, degree of proficiency in the first three languages (for the last two no information is available), early or late onset of bilingualism (before or after age 12) and different language combinations—does not significantly contribute to better performance on measures of cognitive control.

### 3.3. Multilingualism in Usage-Contexts

Following the same procedures as for the static measures above, a linear mixed effects regression models was then fitted for each cognitive measure with the following dynamic ‘usage’ operationalisations of multilingualism: The usage intensity of the first three reported languages across different social domains (family, friends, neighbours and acquaintances) and the degree of contextual switching (whether participants consciously switch languages in a particular context). Because the degree of contextual switching may differ per language per context, we also calculated interaction effects between usage-intensity and contextual switching. Significant results and model summary statistics are presented in Table 7.

The linear regression for the Flanker effect score yielded a significant model with usage intensity of the L2 in different social domains (frequent use of the L2 across different social contexts = slower performance on the Flanker task) and the interaction between L2 usage and contextual switching as significant predictors. This suggests that, when the L2 is used frequently across different social contexts, performance on the Flanker task is in general slower than when the L2 is used less in different contexts. However, in combination with deliberate switching between languages according to specific contextual demands, performance on the Flanker task was significantly faster. AIC model comparisons on the Flanker model with and without the interaction reveals that the model with the interaction performs slightly better (a lower AIC criterion of 3181.1). The interaction effect is plotted in Figure 2 below.

Vice-versa, performance on the WCST was not significantly enhanced (a reduction in number of persistent errors) for the second language and code-switching in interaction. As the total model did not reach significance, the moderate significance of across-domain usage of the L2 and contextual switching in isolation can be discarded. The AIC of the model for the WCST without the interaction was actually slightly lower, yielding a slightly better model (but not significant either: AIC = 1420.2, *R^2^* = 0.011).

### 3.4. PLS Regression Model

We ran a PLS regression model on the variables as outlined in Section 2.4.3. The Flanker model is built with 232 observations, and the WCST model with 220 observations. The crossvalidation results (the root means squared error of prediction, RMSEP, which roughly means the spread of the y values around the regression average) in Table 8 show that retaining 6 components in both models (Flanker and WCST) is enough, but indicate a poor overall fit.

From Table 8 it becomes clear that both PLS regression models have low predictive value. 

To examine which variables contribute most meaningfully we examined the loading weights of each variable, which gives an overview of which variabels load most heavily onto the components and explain most of the variance in the two models (see Table 9 and Table 10. To make this process more insightful, we plot a bar graph of the Variable Importance in Prediction scores (VIP) of all variables for each model. Variables with a score above 1 can be considered important. Figure 3 and Figure 4 show how much each variable contributes to the two models. 

The two VIP plots show that a number of variables importantly contribute to the model. It is thus interesting to examine those variables. 

Because of the overall poor fit, the majority of the loading values are low. Considering the VIP plots and the loading scores, we report the variables that load on component 1 and component 2 above or close to (−)0.2 (rounded), as these seem to be the peaks in the plots. 

Considering the loadings of the variables on the individual components in the models, it becomes apparent that variables relevant in the the first two components cannot be neatly categorised into a specific dimension (e.g., health, or language use). However, those factors in the first component of the Flanker model that are above 0.2 demonstrate that a smaller flanker effect score is positively related to those individuals who are open to new experiences and who use particularly their L2 in different social domains, which explains the most of the variance above all other factors in this component. Why these variables are clustered together is explored in the discussion section. Vice-versa, being less extravert and a limited proficiency in the L2 and L3 are negatively related to flanker performance in the second component. 

Similarly, for the WCST model, more errors are related to lower educational and income levels, as well as being less open to new experiences as a character trait. Higher quality of life and a positive attitude to the L3 reflect fewer errors. The question can, however, be asked to which degree these observations hold.

Nonetheless, it is interesting that beside descriptive predictors such as education and income, outlook on life, either in terms of wellbeing or personality traits, as well as language usage to a small degree influences cognitive performance. We will return to these observations in the discussion.

In the next section, we interpret the findings of the different models in relation to the questions asked in the background section.

## 4. Discussion

The current study analysed the performance of a diverse group of older adults with varying levels of multilingualism on two cognitive tasks relating to (most strongly) inhibition and attention (Flanker) and set-shifting (WCST). The novelty in this analysis lies in the fact that we assessed multilingualism along a continuum and with a dynamic ‘language usage’ operationalisation of multilingualism. In combination with information of factors that are known to enhance cognitive performance from previous studies (such as level of education, and playing a musical instrument) but also other health and lifestyle factors, we built a linear regression model in which we found that degree of contextual second language (L2) usage significantly impacted cognitive performance, but only in one cognitive task and not in another. Purely knowing different languages did not relate to enhanced performance, suggesting that it is not the ability to speak multiple languages, but the use of these different languages that may show small positive effects on cognition. A subsequent multivariate PLS regression model including background variables, language variables from the LEAP-Q, and personality questionnaire factors yielded a two-component solution where proficiency in second and third language and the usage of especially the second and third language across different social domains were predictive of Flanker performance. This underscores the earlier linear regression model mentioned above on the operationalisation of multilingualism, in which language usage variables are more predictive of cognitive advantages than merely ‘knowing’ different languages. We suggest that simply asking a person whether or not they are multilingual is not useful, but probing this more carefully and taking into account proficiency and social language usage patterns does appear to contribute to performance above and beyond the contribution of age, gender, education and income. In the following paragraphs, we critically review our results and answer the two questions that were asked in the beginning of this paper, thereby highlighting possible caveats and misinterpretations of the data, and listing the limitations of our research.

In line with our hypothesis, yet contrary to previous research by [20,21,62], the regression models in Table 6 demonstrated that a traditional knowledge-based operationalisation of multilingualism (as number of languages, early vs. late onset of acquisition, language proficiency and type of language combinations) does not unequivocally lead to enhanced cognitive performance. As individuals differ on many levels, finding an effect of number of languages or degree of cognitive performance would have been strong evidence in favour of a general bilingual advantage. However, given the diverse nature of the study population, plus the uns evidence of a BA reported in the literature, it would have been highly unlikely that in such a heterogeneous population effects for such general observations of multilingualism would be found. As such, our findings are in line with a meta-analysis by Paap et al. [9] of the positive effects of bilingualism on EF (which mostly emerge from small, underpowered studies), where the authors conclude that there are no systematic differences between bi- and monolinguals when regarding these generic factors (early/late, balanced/unbalanced). Indeed, as becomes evident in the PLS regression, there are other, individually distinct factors that covary with measures of multilingualism, such as certain personality traits and higher levels of education.

It may be that so-called ‘confounding variables’ mask the effect of bilingualism on EF (see [36]), or, rather more likely when observing the small effect sizes, help in detecting an effect of language. Perhaps, as some authors suggest, it is by virtue of a number of factors which covary with bilingualism that enhanced EF for multilinguals may be observed. Previous research has attested that factors such as level of education or immigrant status (the ‘healthy migrant effect’) enhance bilingual performance, especially when set-off against a lower-educated or non-immigrant monolingual group. Our PLS regression model demonstrated that multilingualism indeed covaries with a number of other experiential factors in explaining enhanced cognitive performance. When consider the first two components of the Flanker PLS regression, a high quality of life, character traits relating to being open to new experiences and degree of switching languages according to the context covary with the degree of usage of the L2 across different social domains. Proficiency in the second and third language co-occur with extravertness and agreeableness. Together they explain almost 10% of the variance in the Flanker model, which is marginal but noteworthy in a model with such a diverse collection of variables. Although we cannot speculate on the size of the social network of the individuals, scoring high on openness to new experiences might well be an indication of the presence and positive values of social relationships and engaging in social networks. Perhaps those people who use their L2 also in different social domains display a more diverse social network, helped established in part by their personality traits and positive outlook on life. Higher quality of life might imply that people are relatively mobile and/or can more easily maintain their social network. Moreover, the high score of education might reflect a better cognitive disposition from the outset, as enhanced executive functions are often found to be related to educational level.

This is related to the second component, where we observe covariance of L2 proficiency level, age of onset of acquisition of the L1 and extravertness. The contrast between the two components explain different mechanisms of enhanced cognitive performance. Individuals may use their experiences with monitoring language cues, which enhances their attentional control system, through using languages across different social domains. Alternatively, their extensive training in developing language proficiency aids the degree of inhibiting the non-target language. Contrastively, the number of languages one speaks has very low predictive power. This underscores the contextual nature of multilingualism; language use and proficiency, in relation to quality of life and personality variables predicts cognitive performance. This lines up with the argument that other experiential factors may in combination with bilingualism collectively contribute to enhanced cognitive performance [37], but differently in different circumstances. 

Of course there are constraints on the extent to which these variables do actually contribute to enhanced cognitive performance. Moreover, we have presumably tapped into a very specific population sample. The descriptive statistics display that the group was on average highly educated, had a relatively high income, experienced high qualities of life and many of them were early bilinguals. This underscores Bak’s [33] argument that taking note of the environment in which experiments are conducted is vital. As participation was voluntary, we have perhaps attracted mostly those people who already find this type of research topic interesting. (We will come back to the issue of self-selection bias below).

Although our sample consisted of non-immigrant, native-born northern Dutch multilingual older adults, the relative homogeneity of the sample at face-value is distorted when looking more closely at the characteristics of not only health status, wellbeing or personality, but especially degree, type and intensity of multilingualism.

In such a vein, we find differences in the dynamic operationalisation of multilingualism, as degree and intensity of multilingual language usage. What is more, these differences do highlight enhanced performance for some multilinguals on one cognitive task (the Flanker). This result points to the notion that multilingualism is an individually distinct/varying concept, and converges with the observation of those studies that find advantages only for specific populations of bi-/multilinguals or under specific conditions. In our study, especially for the Flanker effect score, those participants who report to use more than one language across different social contexts (with family, friends, neighbours and acquaintances) and especially the L2, and who furthermore switch between languages depending on the social context, show smaller Flanker effect scores. This suggests faster performance on the incongruent trials in comparison to those participants who have a clear usage-preference for one language and/or do not switch in different contexts.

This observation aligns with the research on language balance (for bimodal bilinguals in [27], Yow et al. [63] in younger populations and Houtzager [64] in older populations) and may be explained by the adaptive control hypothesis in that more intense usage of different languages, especially in a dual-language context where bilinguals constantly monitor language cues (and thus focus attention), confers cognitive benefits. It is especially this dual-language context, rather than a dense code-switching context, that incurs benefits in our sample. After all, it is only in interaction with contextual switching that the use of the second language in different social domains leads to faster Flanker performance (in our linear regression model). Without this interaction, the use of the second language across domains yields higher Flanker effect scores, suggesting that there needs to be some element of control/monitoring of attention to language cues that is present in this dual-language mode and which carries over into more general cognitive processes. That the linear model on the WCST error score with the same factors does not reach significance, may give us some insight into the cognitive processes that are at work for this group of multilinguals.

In the WCST, participants have to actively inhibit the old rule in favour of the new one. This switching and inhibition mechanism would, in our understanding of the EF involved in a BA, elicit more accurate performance for those bilinguals who frequently use different languages in different contexts. The absence of an effect here is extra evidence that it is not inhibition per se that drives a potential bilingual advantage, but rather more general attention-orienting behaviour. Bilinguals who use their languages frequently across different social contexts are thus not necessarily better switchers or inhibitors, but may demonstrate an enhanced attention-orienting mechanism.

Evidence for this comes from a study towards the mechanisms underlying the Flanker task, conducted by Ong et al. [65], using diffusion modeling. With this technique, they calculated whether faster response times on the Flanker items are the result of suppressing conflicting information or enhanced attentional control. They found that the bilingual group of older adults showed shorter non-decision times than the monolingual group on incongruent Flanker items, which suggests an enhanced processing efficiency when faced with distracting information. Both groups performed on a par on the other diffusion modeling measures which relate to inhibition, which led the authors to conclude that a BA on the Flanker task is the result of an enhanced attentional control.

Recent theoretical re-evaluations of the processes at work during bilingual decision-making too suggest that the advantage for bilinguals may be in the more general domain of attention selection [3]. Underlying the selection of the appropriate language is the constant monitoring of conflict, which may be part of a more general, non-verbal attention or selection system, Bialystok argues ([3], p. 235). Nonetheless, the WCST also involves focusing attention, yet not in the immediate presence of ‘noise’ or distracting information. Cautious interpretation of the found results is thus warranted as long as we are unclear on what the cognitive tasks actually tap into (also see below).

Moreover, the interaction effect may also be observed as a result of the specific linguistic environment. Recall the study on bilinguals in French-dominant Quebec and English-French bilingual Ottawa, where the linguistic environment proved to be a more robust explanation for enhanced executive control than bilingualism in itself [32]. As exposure to different languages in our group is also relatively high—the Frisian language and Groninger and Drents dialects are omnipresent in the provinces—it may well be that we observe not an effect of social language switching, but simply an effect that culminates from the ability to use the different languages in the immediate environment. This observation underscores the importance of reporting on the social norms and status of different languages in populations, as pointed out by Surrain and Luk [34].

The PLS regression that we computed to answer the question of whether multilingualism could have any explanatory power regarding better cognitive performance iterates the linear model on the operationalisation of multilingualism in that more crude measures such as number of languages do not predict much. Rather, more sensitives measures relating to language usage and proficiency combine with wellbeing and personality traits, and are together informative of cognitive performance. The PLS regression technique limits the degree of ‘gold-digging’ in the data to find patterns that in hindsight can be explained by existing theories, something that Hartsuiker rightfully warns for [13]. Nonetheless, even with such a diverse sample of variables the overall explanatory power of both models is pretty low. This underscores the importance to establish a clear theory of the relation between language and cognitive performance. Our findings regarding the operationalisations of multilingualism above may be especially enlightening for this purpose when considering how language control modulates cognitive control.

Beyond this finding, however, there are still gaps in the knowledge pertaining to exactly how language engages executive functions. In line with De Bruin and Della-Sala [66], we advocate that comparing groups of bi- and monolinguals should not form the foundation of such a framework, but rather the characteristics of bilingualism, especially language usage patterns, following the adaptive control hypothesis. However, we are still unsure as to how this task-specific executive control (either inhibition, attention direction, or some other mechanism) transfers to more broader domains of cognitive control, and whether a Flanker task and a WCST aptly measure these control processes (also see [67]). Nonetheless, the observation that multilinguals who use their languages often and in different contexts demonstrate slightly enhanced performance on the Flanker task, but not the WCST suggests that language use, or the multilingual linguistic environment are contributing factors in enhanced attention-direction. Whether this transfers to other executive functions or cognitive domains cannot be ascertained, given the overall low significance of the linear regression models.

### Limitations

There are a number of other explanations for the found (absence of) effects of multilingualism on EF that warrant a cautious interpretation of the found results. The absence of an effect of the ‘knowledge’ variables associated with multilingualism may have emerged because the majority of the participants report the maximum number of five languages they could list. These are not only the languages and/or dialects they use frequently, but also the languages that they have learned in school but which they do not use as productively. This results in a population that is already highly multilingual in terms of number of languages. Self-reported proficiency may also have biased the data. In addition, it is likely that proficiency in language 2 and 3 is higher in general than in an average population, as those people who responded to the questionnaire may well be ‘advanced’ speakers of multiple languages, resulting in a self-selection bias (also see below).

Given the subjectivity with which we measure proficiency we do not rule out an effect of proficiency found in other studies. However, measuring proficiency is notoriously difficult and interpretation in degree of bi- or multilingualism is inherently flawed [68]. Can there be a ‘cut off’ point at which someone with a certain level of proficiency is considered more multilingual than others? The absence of an effect of proficiency in this study, we therefore argue, can most solidly be explained by the lack of variation/differentiation in proficiency scores.

Self-reports come with inherent shortcomings. Participants may be more or less optimistic in judging their language abilities, also depending on their personal motivation to participate in the study. A self-selected test population is always biased. The majority of the participants likely participated because they consider themselves multilingual language users and have a positive attitude towards using their languages (which we explicitly asked in the questionnaire). They may be proud speakers of minority languages and/or dialects. By approaching multilingualism from an ‘inclusive’ perspective, meaning that it is more important whether someone speaks/uses multiple languages than their relative degree of proficiency in these languages, some participants may have listed their knowledge of languages in which they have a very basic proficiency and which they sparsely use, whereas others have listed only those languages which they frequently use. These differential interpretations we have tried to keep in check by being as detailed as possible in our questionnaire and asking not only questions on knowledge of languages but also usage patterns. Furthermore, as referenced in the materials section, participants completed the questionnaire in an online environment, which already selects those participants with computer skills. A small minority of the participants completed the questionnaire by means of a face-to-face interview. Because we have tried to match the conditions of the interview procedure as closely as possible with the online questionnaire, and because the ‘interview participant’ sample was so small, we do not expect any qualitative differences in the results to occur because of these different modes of data collection. Nonetheless, the reliability of the data can only be warranted by performing replication studies with different populations.

## 5. Conclusions

Multilingualism is a life-experience that, when different languages are used frequently in different contexts, can contribute to enhanced attention control. In a larger model with multiple experiences added, not the number of languages but the degree of proficiency in especially the L2 and L3 and the use of the L2 across different social contexts are predictive of cognitive effects. This strengthens the argument that multilingualism may be one of the contributing factors to enhanced cognitive performance, but the strength of contribution varies per individual and covaries with personality and wellbeing measures, as well as with education. We observed these effects in a very specific population of (motivated) older multilinguals with diverse language/dialect backgrounds. Future research should replicate this study with other bilingual populations.

This study crucially contributes to the debate surrounding BA’s, by highlighting that we should move away from knowledge-based operationalisations of multilingualism and conduct more fine-grained, individual analyses based on language usage. After all, language is first and foremost a vehicle for communication, something which always happens in context and in interaction. Singling out language effects in carefully controlled experiments is therefore likely to yield artificial results that cannot be translated to real-life language use contexts.

## Figures and Tables

**Figure 1 brainsci-08-00092-f001:**
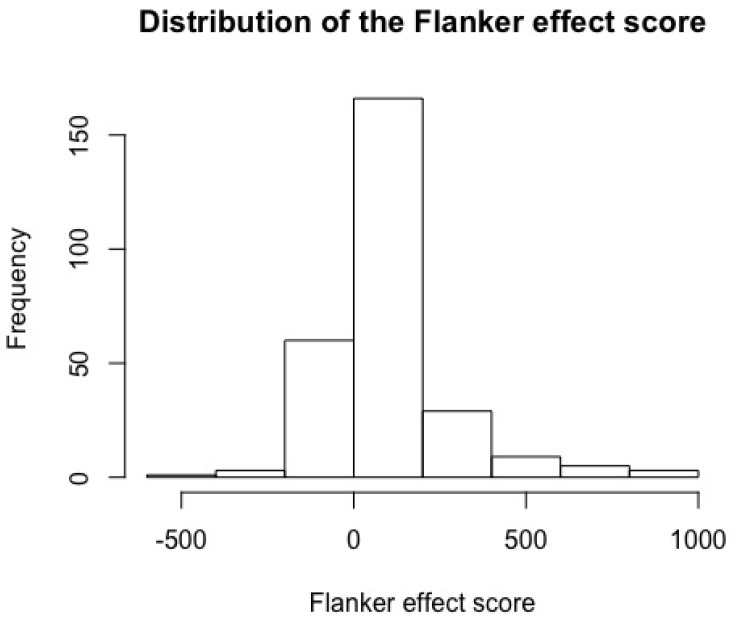
Distribution of the Flanker effect score.

**Figure 2 brainsci-08-00092-f002:**
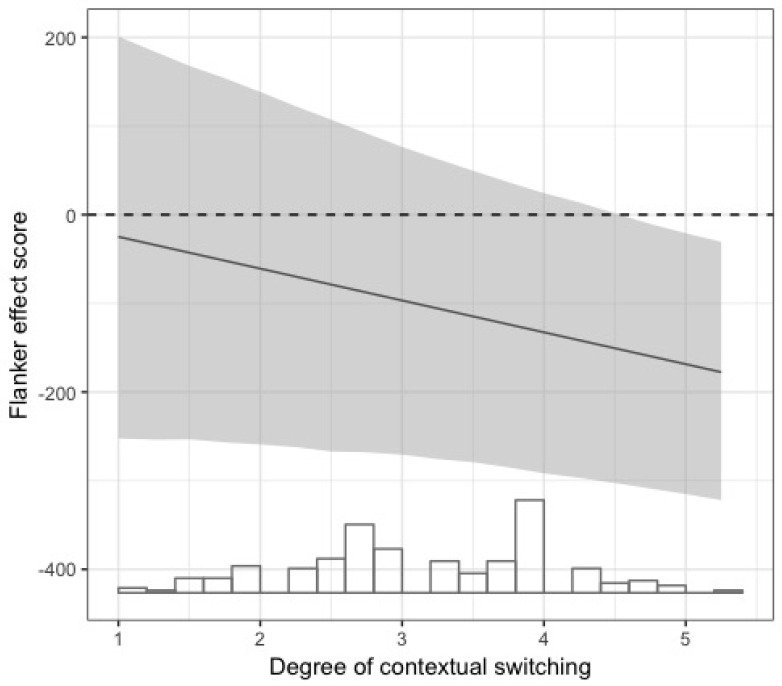
Estimated coefficient of the Flanker effect score versus degree of contextual switching by across-domain usage of the L2.

**Figure 3 brainsci-08-00092-f003:**
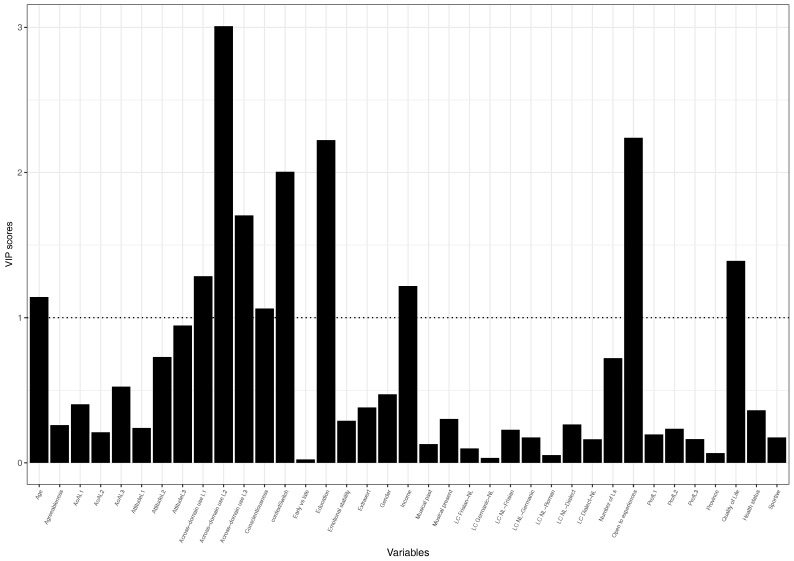
VIP plot of Flanker PLS model.

**Figure 4 brainsci-08-00092-f004:**
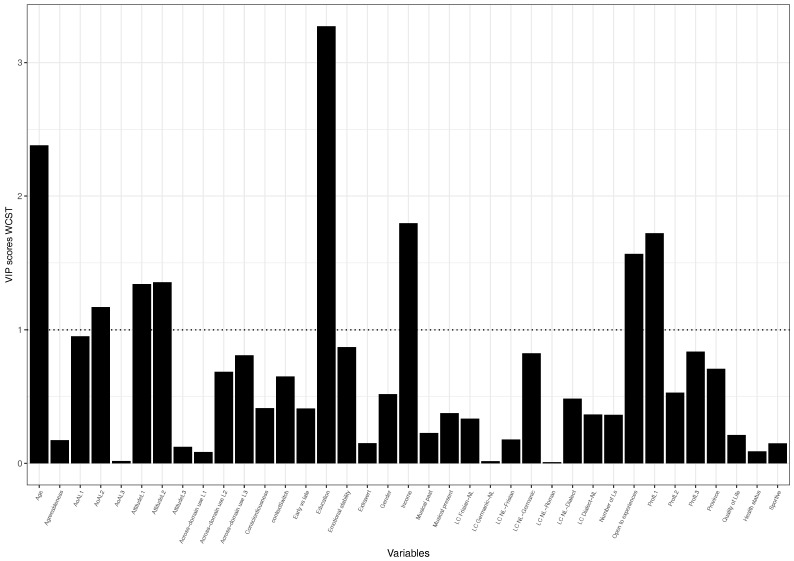
VIP plot of WCST PLS model.

**Table 1 brainsci-08-00092-t001:** Demographics of the participant sample.

Statistic	*N*		Mean		St. Dev.	Min	Max
Age	387		72.067		5.708	65	95
Gender	Male	201	-		-	-	-
	Female	185					
Province	384		Friesland	173	0.840	-	-
			Groningen	103			
			Drenthe	108			
Education	387		4.925		1.073	2	6
Income	387		6.866		1.400	3	9
Self-reported health	386		3.311		0.924	1	5
Multimorbidity	387		1.388		1.271	0	7
QoL	387		8		0.908	4	10
Sport	387		Yes	282	-	-	-
			No	105			
Playing an instrument	387		No	261	-	-	-
			Yes, passive	66			
			Yes, active	60			

**Table 2 brainsci-08-00092-t002:** Summary of outcomes of the language measures.

Statistic	*N*		Mean	St. Dev.	Min	Max
Number of languages	387		4.199	1.002	1	5
Proficiency L1	371		4.881	0.381	1	5
Proficiency L2	365		4.565	0.613	1	5
Proficiency L3	341		3.898	0.808	1	5
AoA L1	376		3.148	3.504	0	24
AoA L2	368		7.649	8.698	0	68
AoA L3	340		13.532	9.146	0	67
Early or late acquisition of L2	Early	264	-	-	-	-
	Late	104				
Positive attitude L1	378		4.587	0.690	2	5
Positive attitude L2	372		4.315	0.831	1	5
Positive attitude L3	345		4.020	0.787	1	5
Across-domain L1	383		4.163	0.833	1.000	5.000
Across-domain L2	377		3.139	0.956	1.000	5.250
Across-domain L3	343		1.897	0.781	0.250	4.500
Degree of contextual switching	365		2.450	0.791	1.000	4.670

**Table 3 brainsci-08-00092-t003:** Items in the background questionnaire.

Theme	Adapted From	Items
Demographics	TOPICS-MDS [42]	Age (in years) Province of residence (1 = Friesland, 2 = Groningen, 3 = Drenthe) Place of birth Education (6-point scale; 1 = only primary school to 6 = University degree) Income (9-point scale; 1 = 500 with increments of 500 until 9 => 3000) Hobbies
Health information	TOPICS-MDS	Self-reported health (scale measures, see below) Multimorbidity Functional health Resilience
Quality of life	TOPICS-MDS	Self-reported quality of life (mark between 1 (low) and 10 (high)) Emotional wellbeing
Language knowledge	Not applicable	Number of languages and which, according to dominance and order of acquisition Use of each language in past two weeks Use of each language in different social domains
Language usage (x3)	LEAP-Q [16]	Age of onset of acquisition Degree of proficiency (speaking/reading/writing/comprehension) Mode of acquisition Reading/TV/radio/internet usage in each language Attitude toward each language
Switch behaviour	Bilingual Language Switch Questionnaire [41]	Degree of contextual switching Degree of control over switching Degree of conscious switching
Personality	TIPI questionnaire [43]	Ten questions targeting the ‘Big Five’ personality traits: extraversion, conscientiousness, openness to new experiences, agreeableness, emotional stability.

**Table 4 brainsci-08-00092-t004:** Overview of cognitive tests.

Cognitive Test	Reference	Cognitive Process	Outcome Measure
Flanker	[55]	Attention, inhibition	Flanker effect score in ms
WCST	[56]	Switching, set-shifting	Total number of persistent errors
Corsi blocks	[57]	Working memory	Total forward span

**Table 5 brainsci-08-00092-t005:** Overview of cognitive task performance.

Statistic	N	Mean	St. Dev.	Min	Max
Flanker effect score	276	95.62	174.36	−550.6	895.85
WCST error score	258	12.79	4.93	4	29

**Table 6 brainsci-08-00092-t006:** Summary statistics of the two linear mixed effects regression models on the outcomes of the two cognitive tasks with a static intepretation of multilingualism.

	Dependent Variable	
	Flanker Effect Score	Persistent Errors
	(Beta Weights and SEs)	(Beta Weights and SEs)
Observations	244	231
*R* ^2^	0.043	0.057
Adjusted *R*^2^	−0.006	0.005
Residual Std. Error	165.608 (*df* = 231)	4.945 (*df* = 218)
F Statistic	0.875 (*df* = 12; 231)	1.103 (*df* = 12; 218)

**Table 7 brainsci-08-00092-t007:** Mulitple linear regression models of cognitive performance related to dynamic operationalisations of multilingualism.

	Dependent Variable	
	Flanker Effect Score	Persistent Errors
	(Beta Weights and SEs)	(Beta Weights and SEs)
Across-domain use L1	−25.673	2.693 *
	(47.527)	(1.612)
Across-domain use L2	119.110 ***	2.342 *
	(40.512)	(1.330)
Contextual switching	−15.704	6.723 *
	(109.204)	(3.655)
Use L2:CS	−36.873 **	−0.755
	(15.869)	(0.521)
Constant	−42.054	−7.482
	(287.928)	(9.715)
Observations	246	234
*R* ^2^	0.106	0.028
Adjusted *R*^2^	0.080	−0.002
Residual Std. Error	152.415 (*df* = 238)	4.947 (*df* = 226)
F Statistic	4.041 *** (*df* = 7; 238)	0.937 (*df* = 7; 226)

Note: * *p* < 0.1; ** *p* < 0.05; *** *p* < 0.01.

**Table 8 brainsci-08-00092-t008:** Multiple linear regression models with only significant effects reported for demographic, health, language and personality factors.

	Dependent Variable	
	Flanker Effect Score	WCST Persistent Errors
	(6 Components)	(6 Components)
CV value	169.8	5.429
% of explained variance	12.83	19.46

**Table 9 brainsci-08-00092-t009:** Loading values above (−)0.2 of Flanker PLS regression model.

	Flanker Effect Score	
	Component 1	Component 2
	(6.5% Variance)	(7.1% Variance)
Education	−0.365	
Income	−0.200	
QoL	0.228	
Contextual switching	0.330	
Open to experiences	0.368	
Across−domain L1	−0.211	
Across−domain L2	0.495	
Across−domain L3	0.280	
Proficiency L2		−0.373
Proficiency L3		−.252
AoA L1		−0.247
Extravertness		−0.283
Agreeableness		−0.303
Province of residence		0.229

**Table 10 brainsci-08-00092-t010:** Loading values above (−)0.3 of WCST PLS regression model.

	WCST Error Score	
	Component 1	Component 2
	(7.4% Variance)	(6.5% Variance)
Age	0.391	
Education	−0.538	
Income	−0.295	
Proficiency L1	−0.283	
Open to experiences	−0.258	
QoL		0.437
Attitude L3		0.436
Emotional stability		0.280

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
