# Peer review of "Intensity of Multilingual Language Use Predicts Cognitive Performance in Some Multilingual Older Adults"

_brainsci, 2018, doi:10.3390/brainsci8050092_

Round 1

Reviewer 1 Report

Comments on Pot, Keijzer, & de Bot: Intensity of multilingual language use predicts cognitive
performance in multilingual older adults
This paper reports an ambitious study on the cognitive consequences of multilingualism for older
adults in the Netherlands. The Netherlands is a multilingual country, with most individuals more
than bilingual, having knowledge of up to four or five languages. The approach taken in the
present study was to focus on variation within this group of older adult speakers rather than to
compare bilinguals and monolinguals as a means to assess the contribution of language
experience to performance on executive function tasks. The recent literature on the consequences
of bi and multilingualism has been controversial. Some studies have failed to replicate earlier
reported advantages of bilingualism or multilingualism on measures of executive function,
particularly in young adult bilinguals, and particularly in behavioral assessments of cognitive
performance. Those failures to replicate have raised questions for some about whether there is
any relation between the use of more than one language and cognitive function.
A problem in the past literature is that much of the previous research fails to acknowledge the
complexity of multilingualism and fails to consider the ways that two or more languages are used.
The present paper represents an approach that is beginning to be taken more broadly and that
embraces the variation in language use and language contexts to begin to identify the
circumstances under which specific types of language experiences hold consequences for
cognitive performance. This is an enormously important and difficult task because, as the authors
note, even in a location where multilingualism is the norm, there are many differences among
speakers and contexts that may contribute to the observed outcomes.
The design of this study was to sample a very large group of older Dutch adults, all of whom are
multilingual and to determine which aspects of their language and life experience contribute to
cognitive performance on two widely used measures of executive function: the flanker task and
the Wisconsin Card Sorting task. The approach was to harness the power of a large sample and
regression statistics to be able to tease apart the potentially subtle effects of differences in
language use. Two results in this work are critical. First, they find an effect of what they call
contextual switching, the use of different languages in different contexts in a dual language
environment. Second, the enhancement to cognitive performance is found only for the flanker
task, not for the Wisconsin Card Sorting task. In light of the controversial discussion about these
consequences of bilingualism these results are important because different measures of executive
function are likely to reflect the engagement of different components of cognitive control. In a
larger statistical model with many other factors, multilingualism makes a modest contribution
only.
What is the take away message? The results of this study show that benefits of multilingualism
to cognitive function in older adults are not due to speaking more than one language per se, nor to
the age at which the languages were acquired, nor to the presence of language switching per se.
Rather, they appear to be due to the contexts in which speakers are likely to have to monitor their
use of different languages across different settings. The mechanism that underlies the observed
benefit is unlikely to be attributable to the operation of a single or simple cognitive process but to
the way that the networks that support cognitive function are tuned and change dynamically when
individuals must attend to a wide range of variables that determine with whom they can speak
which of their languages. On this view, some traditional comparisons of bilinguals/multilinguals
and monolinguals will show no effects of language experience whereas others will show
significant effects. Unless this complexity associated with the context of language use is taken
into account, the story about these cognitive consequences will be incomplete.
As the authors acknowledge, the study is not perfect. It relies on measures of self report that are
potentially less reliable than objective measures of language performance. The use of the two
executive function tasks is a strength but again, it is only a partial lens into the likely mechanisms
of attention and cognitive control that are affected by language experience. As the authors also
note, despite the variation within the studied sample, the high level of multilingualism in this
group may have had the effect of imposing a ceiling of sorts that may mask the range of
consequences. The main contribution of this work is to show that unless we begin to understand
the features of life experience with more than one language, we will not be able to build a model
of language and its cognitive consequences that is accurate. This is a hard task and this paper
provides an encouraging model for how we can begin to approach this challenge.
This paper will make a very useful contribution to the current literature.
Judith Kroll
April 13, 2018

Reviewer 2 Report

This paper represents a rescue effort of sorts for the field of bilingualism which has
come under heavy criticism of late from the likes of Paap and colleagues. Part of the critique
that has been levelled against the field is, as the authors note, because bilingualism is so
poorly defined. The authors attempt to amend this by examining bilingualism in a more
nuanced way. To wit, they examine whether language use in different contexts (in contrast to
mere knowledge of multiple languages - styled by the authors as “static knowledge”) might be
a more helpful predictor of attention (measured in this case by the Flanker Effect). The authors
run several linear mixed regression models and conclude that a) the static multilingual variables
contain no predictive power for performance on either the Flanker task or the Wisconsin Card-
Sorting Task, this they argue is in line with a recent meta-analysis by Paap et al. Followup
models including use variables yielded significant effects. “Usage intensity of the L2 in
different social domains” led to slower Flanker Effects, while the interaction between L2 usage
and contextual switching actually led to faster Flanker Effects. The authors suggest that their
results provide evidence against the conventional “bilingual advantage” type pieces, thus
supporting Paap and colleagues, and instead provide evidence that a specific type of bilingual
usage score may be a more fruitful avenue of research for future study. I applaud the authors
use of continuous variables, their avoidance of categorizing “bilinguals” and monolinguals, and
their examination of contextual effects on bilingualism (à la Green & Abutalebi’s 2013 adaptive
control hypothesis). However, I still have some concerns which must be addressed before this
paper is published.
On the whole, I like this approach, and I think that this paper should (eventually) be published
as it will help to reorient the field of bilingual research back towards its roots, and this paper
goes a long way to helping restore needed nuance to the field.
Major concerns:
1. The authors state that their data is available online in DataverseNL. While this is a
commendable step, I could not find the data there. Please make the raw data available
before the next revision as I would like to examine your findings. In addition, please make
the R script used to preprocess and analyze the data available with the dataset. Included
scripts should be sufficient to reproduce all the reported results.
2. The idea that a single measure of bilingualism, or even a measure of proficiency is not
sufficient to capture the complexity of the bilingual experience is not a new idea. Indeed in
1965, Robert Cooper noted that Puerto-Rican Spanish and English scores on word and
category generation were affected depending on five “societal domains”, family,
neighborhood, religion, education, and work. Later studies also reemphasized this.
Grosjean (1998) hi lighted the fact that “research dealing with bilinguals has often produced
conflicting results” and that “some researchers, admittedly a few, do not yet fully share the
field’s understanding of who bilinguals really are.” He advocated that more complete
documentation of language history be provided, preferably with a standardized set of
questionnaires, and that language language modes be taken into account. Language
modes in Grosjean’s conceptualization addressed how long a person was in a monolingual
mode, how long they were in a bilingual mode, and when in a bilingual mode, how much
switching is taking place. Similar sentiments are echoed by Luk & Bialystok (2013), in their
paper Bilingualism is not a categorical variable. Thus, the authors are not expressing a
novel idea when they suggest that more complex conceptualizations of bilingualism are
needed, indeed they are perhaps returning the field to its roots.
3. The authors collected a moderate sample of 387 participants. Demographic variables and
results from the language testing must be presented in an accompanying table. Unless this
information is included (per Grosjean’s 1998 suggestion), we do not know anything about
this particular set of bilinguals, nor to what extend these results are generalizable to any
other population.
4. The authors note that 90% of the data were obtained by individuals in their own homes via
an internet survey (Qualtrics, and PsyToolkit). Do the data from the individuals who
received in-home instruction match the data for participants who did the experiment on
their own? In short, does the data quality meet the same standard as that obtained in an
experimental setup?
5. Why did the authors not choose to examine the outcome using a multivariate technique like
PLS regression? I can think of many advantages to this latter approach. First, it can
handle correlated variables, second, the model becomes less about which individual factor
predicts the outcome best, but which combination of variables explains the most
covariance in the outcomes, and finally, both outcome variables could be fit simultaneously.
6. How were the covariates chosen? Including openness to experience is interesting since
one might expect that individuals who choose to become bilingual might be more open -
hence this seems a bit like an attempt to drown the explanatory power of any bilingual
effect by including random covariates. If openness to experience, why not neuroticism?
7. It is clear from a number of studies that bilingualism usually does not explain more than
education in models of cognition or cognitive reserve (e.g., Ihle et al., 2016). Including
education in the model unsurprisingly removes some of the explanatory power of
bilingualism. This is a bit like controlling for intelligence when trying to predict a cognitive
outcome - very often one ends up “throwing the baby out with the bathwater”. More
interesting is what combination of predictors leads to the best outcome. Again, the
multivariate partial-least-squares regression model is well suited to this.
8. The authors chose to trim their data (300-1700 ms). Please a) show the results with and
without the trim and b) ensure that the raw data is available on DataVerse. Any trimming
should be done in the accompanying .R file. The concern here is that, as the authors note,
sometimes effects of bilingualism occur in the tail of reaction time distributions (e.g.,
Abutalebi et al., 2015; Calabria et al., 2011). Given that this part of the distribution has
been shown to differentially affect the outcome variables, excluding it is concerning unless
the authors can show it has no effect.
9. The authors include dialects of languages as well as accents in their accounting of
multilingualism. As far as I know, “bilingual advantage” effects have never been attributed
to accents. Please re-do the analyses without counting dialects and accents separately. If
the results hold, report the results as originally presented with a note to that effect.
Minor concerns:
10. Citation to Kave (page 3 line 81) is incorrect - I think you want to refer to the aging paper,
not the one on children.
11. Lines 163-166 consider breaking this sentence up to aid comprehension.
12. What is being presented in the Tables 4, 5 and 6? Beta weights from the regression with
standard errors? Flanker effects with standard deviations? More information is needed.
13. Figure 2: what are the error bands?

Round 2

Reviewer 2 Report

The revision to this paper has improved.  I particularly appreciate the effort the authors took to include the multivariate PLS model.  That said, I still have concerns that need to be addressed before this paper can be published.  In particular, the interpretation of the PLS results need to be refined and the tone of the paper changed.  It is clear to me from these new analyses, that while number of languages might not be a very good predictor of cognitive performance (and really, this is not very surprising), proficiency in second and third, and comprehension of the second languages are among the most powerful predictors of Flanker performance in this data set.  This needs to be a core focus of the discussion and needs to be underscored in the abstract.  I would write something like the following: "A multivariate PLS regression including background variables, language variables from the LEAP-Q, and personality questionnaire factors yielded a two-component solution where proficiency in second and third language and comprehension of the second language were predictive of flanker performance.  This contrasts with the crude measure of how many languages the person spoke which contributed little to the overall model.  We suggest that simply asking a person whether or not they are bilingual is not useful, but probing this more carefully and taking into account proficiency and comprehension does appear to contribute to performance above and beyond the contribution of age, gender, education, and income."  The authors could add something about how proficiency in second and third language and comprehension of the second language do in fact covary with extraversion, conscientiousness etc...

So, in short, it appears that by including the PLS model, we now see that focusing on the "number of languages" is not very useful.  But one can't cherry pick the single predictor in a multivariate model and choose to talk about that.  All are now fair game.  The authors need to restructure their paper to reflect the fact that if you use a crude measure like number of languages spoken, not much will be predicted.  By contrast, using more sensitive measures of second and third language proficiency and comprehension appear to be highly informative.  This new model and my reading of it appear to contradict the authors conclusions and their interpretations in the abstract.  This will need to be fixed before the paper can be published. 

Major comments:

I previously asked whether there was a difference between the quality of the data obtained via participants online or with an experimenter in their own home.  Am I right in interpreting that the authors did *not* record when this difference occurred?  If so, that is most unfortunate, and a limitation that needs to be made explicit in the manuscript. 

Clearly yes, the number of languages in the PLS analysis do not explain very much in the first latent variable (more on this later), but a number of other language related variables including proficiency contribute in a large way (more than 0.2).  This suggests that proficiency, not merely number of languages is more informative for performance on this task.  Please emphasize this in your results and discussion section.  It is also interesting to note that age of acquisition in the first latent variable does not add much to the model. 

The two components that were extracted in each case from each of the PLS analyses were nearly of equal weight (both close to 7%).  This means that BOTH components need to be afforded close attention in the discussion.  The authors do a fairly good job of describing the results of the first component.  Number of languages do contribute positively, but in a small way to this component (describing Flanker performance) while proficiency is clearly a stronger predictor.  Similar attention must be paid to the second component in the discussion and in the abstract.  This second component is orthogonal to the first suggesting that once one strips away the large effects associated with QoLrapport (whatever this is), proficiency in second and third languages, personality variables, and comprehension of the second language (which interestingly explains more than any other single variable in the PLS model), the second, orthogonal contrast describes the covariance of education and proficiency in the second language. 

I have a few minor comments.

The figures for the PLS analysis needs to be redone, they are not yet publication ready.

First, add a reference line at zero

Add meaningful axis labels & make sure they do not overlap the variable names

format the label names properly

Rather than line-graphs, consider plotting a bar-graph

Consider bootstrapping the model & adding 95% confidence intervals around each of the loading scores.  This would allow the reader to interpret whether the variable contributes reliably more than 0, or from other variables. 

References appear to be missing/appear with question marks throughout.  Please check the bibtex file/recompile the paper before resubmitting.

This manuscript is getting there, but I don't think the data describe what the authors believe it does.  Certainly the PLS model provides evidence that second and third language variables contribute far more than is described by the authors, and their attempts to dismiss these findings are misleading. 

Author Response

To the reviewer of our manuscript and the editor of Brain Sciences,

In this second round of reviewing our manuscript, we would sincerely like to thank the reviewer for the great amount of time invested in the thorough reading of our manuscript and the many useful suggestions and feedback. This is highly appreciated.

We will again respond to the issues raised by the reviewer (in italics) point-by-point, and have highlighted the modified parts in the manuscript in red.

The revision to this paper has improved.  I particularly appreciate the effort the authors took to include the multivariate PLS model.  That said, I still have concerns that need to be addressed before this paper can be published.  In particular, the interpretation of the PLS results need to be refined and the tone of the paper changed.  It is clear to me from these new analyses, that while number of languages might not be a very good predictor of cognitive performance (and really, this is not very surprising), proficiency in second and third, and comprehension of the second languages are among the most powerful predictors of Flanker performance in this data set.  This needs to be a core focus of the discussion and needs to be underscored in the abstract.  I would write something like the following: "A multivariate PLS regression including background variables, language variables from the LEAP-Q, and personality questionnaire factors yielded a two-component solution where proficiency in second and third language and comprehension of the second language were predictive of flanker performance.  This contrasts with the crude measure of how many languages the person spoke which contributed little to the overall model.  We suggest that simply asking a person whether or not they are bilingual is not useful, but probing this more carefully and taking into account proficiency and comprehension does appear to contribute to performance above and beyond the contribution of age, gender, education, and income."  The authors could add something about how proficiency in second and third language and comprehension of the second language do in fact covary with extraversion, conscientiousness etc... 

So, in short, it appears that by including the PLS model, we now see that focusing on the "number of languages" is not very useful.  But one can't cherry pick the single predictor in a multivariate model and choose to talk about that.  All are now fair game.  The authors need to restructure their paper to reflect the fact that if you use a crude measure like number of languages spoken, not much will be predicted.  By contrast, using more sensitive measures of second and third language proficiency and comprehension appear to be highly informative.  This new model and my reading of it appear to contradict the authors conclusions and their interpretations in the abstract.  This will need to be fixed before the paper can be published.  

We would first of all like to comment on this overall statement regarding the reading of the PLS regression model. We greatly appreciate the suggestions by the reviewer. Our experience with PLS regression is thus far limited to the analyses of the data that form the basis of this manuscript, and we value the reading of the results by the reviewer. However, there seems to have occurred some misunderstanding as to the interpretation of these results.

We have not assessed language comprehension in our models. We believe that the misunderstanding has arisen due to the reading of the variables “complang1/2/3” which does not denote comprehension but “composite score” for the domains of language use. The variable has been rather inaptly named we now realise, and has caused confusion in the interpretation of the results; where we talk about degree of language usage across different social domains (complang1, complang2 and complang3) in relation to cognitive performance. Admittedly, we have not discussed the large contribution language proficiency makes in the model, and this is something that needs to be stressed much more, as the reviewer rightly points out.

To make sure that we are on the same page regarding the interpretation of the results, we would like to build upon the paragraph suggestion the reviewer wrote and write the following:

“A multivariate PLS regression including background variables, language variables from the LEAP-Q, and personality questionnaire factors yielded a two-component solution where proficiency in second and third language and the usage of especially the second and third language across different social domains were predictive of Flanker performance. This underscores a linear regression model on the operationalisation of multilingualism, in which language usage variables are more predictive of cognitive advantages than merely ‘knowing’ different languages. We suggest that simply asking a person whether or not they are multilingual is not useful, but probing this more carefully and taking into account proficiency and social language usage patterns does appear to contribute to performance above and beyond the contribution of age, gender, education and income”.

Major comments:

I previously asked whether there was a difference between the quality of the data obtained via participants online or with an experimenter in their own home.  Am I right in interpreting that the authors did *not* record when this difference occurred?  If so, that is most unfortunate, and a limitation that needs to be made explicit in the manuscript. 

We did indeed not record whether participants filled in the questionnaire online or via a face-to-face interview. However, as we made sure that the conditions for the interview match the online questionnaire procedure as closely as possible, we do not expect any qualitative differences between the two modes of data collection. Moreover, the number of persons participating through a face-to-face interview was very low (n=13), which makes it highly unlikely that any findings in the models could be ascribed to the two different methods of data collection. However, we have made this fact more explicit in the limitation section of the manuscript on page 24 as follows:

“Furthermore, as referenced in the materials section, participants completed the questionnaire in an online environment, which already selects those participants with computer skills. A small minority of the participants completed the questionnaire by means of a face-to-face interview. Because we have tried to match the conditions of the interview procedure as closely as possible with the online questionnaire, and because the ‘interview participant’ sample was so small, we do not expect any qualitative differences in the results to occur because of these different modes of data collection.” (page 24-25, line number 783).

Clearly yes, the number of languages in the PLS analysis do not explain very much in the first latent variable (more on this later), but a number of other language related variables including proficiency contribute in a large way (more than 0.2).  This suggests that proficiency, not merely number of languages is more informative for performance on this task.  Please emphasize this in your results and discussion section.  It is also interesting to note that age of acquisition in the first latent variable does not add much to the model.  

The two components that were extracted in each case from each of the PLS analyses were nearly of equal weight (both close to 7%).  This means that BOTH components need to be afforded close attention in the discussion.  The authors do a fairly good job of describing the results of the first component.  Number of languages do contribute positively, but in a small way to this component (describing Flanker performance) while proficiency is clearly a stronger predictor.  Similar attention must be paid to the second component in the discussion and in the abstract.  This second component is orthogonal to the first suggesting that once one strips away the large effects associated with QoLrapport (whatever this is), proficiency in second and third languages, personality variables, and comprehension of the second language (which interestingly explains more than any other single variable in the PLS model), the second, orthogonal contrast describes the covariance of education and proficiency in the second language.  

In line with the overall comment above, we have restructured parts of our discussion to reflect this outcome better. We have added an explanation also for the second component. Please refer to the highlighted parts in the discussion on page 22, particularly from line 696 onwards.

Also, in the previous version we split the data in a train and test set. However, as we now bootstrapped the model we have redone the analyses with the complete dataset. Although the overall pattern remained the same, there are some slight differences in the scores (loading weights) of each variable, which we have adapted in the tables on page 18-19.

I have a few minor comments.

The figures for the PLS analysis needs to be redone, they are not yet publication ready.

First, add a reference line at zero

Add meaningful axis labels & make sure they do not overlap the variable names

format the label names properly

Rather than line-graphs, consider plotting a bar-graph

Consider bootstrapping the model & adding 95% confidence intervals around each of the loading scores.  This would allow the reader to interpret whether the variable contributes reliably more than 0, or from other variables.

After reconsidering the graphic representation of the loading variables, we have opted to include a Variable Importance in Prediction plot for the first two components of each model. This plot renders a better interpretability of the relative importance of each variable to the overall model. Variables with a score above 1 can be considered important. We believe that this allows the reader to easily assess which variables contribute to the model (more so than in the loadings plot).

Because of the inclusion of the VIP plot, we do not provide bootstrapped confidence intervals for the variable loadings. However, we have computed them in the accompanying .R file.

We have formatted the label names properly so that they now accurately describe what is measured.

References appear to be missing/appear with question marks throughout.  Please check the bibtex file/recompile the paper before resubmitting.

Thank you for observing this. It might have to do with the PDF compiler used by the Brain Sciences submission system, because when we compile the PDF there are no missing references/questionmarks. To be on the safe side, we will upload a compiled PDF with the resubmitted manuscript.

This manuscript is getting there, but I don't think the data describe what the authors believe it does.  Certainly the PLS model provides evidence that second and third language variables contribute far more than is described by the authors, and their attempts to dismiss these findings are misleading.

We would once again like to underscore the relevance of the suggestion regarding the usage of the PLS model. It has indeed greatly improved the manuscript because it allows for a more fine-grained look at the contributions of the different variables. We believe that our manuscript can, now more than in its first version, contribute to the field of bilingual advantages by offering insights into some of the contributing (and covarying!) factors involved in multilingual cognitive aging.

Kind regards,

The authors
